# Factors associated with retention in Quitline counseling for smoking cessation among HIV-positive smokers receiving care at HIV outpatient clinics in Vietnam

Nam Truong Nguyen[1]*, Trang Nguyen[1], Giap Van Vu[2,3], Charles M. Cleland[4], Yen Pham[1], Nga Truong[1], Reet Kapur[5], Gloria Guevara Alvarez[5], Phuong Thu Phan[2,3], Mari Armstrong-Hough[5], Donna Shelley[5]

1 Institute of Social and Medical Studies, Ha Noi, Vietnam, 2 Bach Mai Hospital, Ha Noi, Vietnam, 3 Ha Noi Medical University, Ha Noi, Vietnam, 4 New York University Grossman School of Medicine, New York, NY, United States of America, 5 School of Global Public Health, New York University, New York, NY, United States of America

* ntnam@isms.org.vn

**Data Availability Statement:** All relevant data are within the manuscript and its Supporting Information files.

## Abstract

### Background

Quitline counseling is an effective method for supporting smoking cessation, offering personalized and accessible assistance. Tobacco use is a significant public health issue among people living with HIV. In Vietnam, over 50% of men living with HIV use tobacco. However, there is limited research on Quitline use and retention rates in this population and a lack of research on factors associated with retention in Quitline counseling. The study aims to evaluate the factors associated with retention in Quitline counseling for smoking cessation among HIV-positive smokers receiving care at HIV outpatient clinics in Vietnam.

### Method

The study analyzed data from a randomized controlled trial (RCT) that compared the effectiveness of three smoking cessation interventions for smokers living with HIV at 13 Outpatient Clinics in Ha Noi. A total of 221 smokers aged 18 and above living with HIV participated in Arm 1 of the RCT, which included screening for tobacco use (Ask), health worker-delivered brief counseling (Assist), and proactive referral to Vietnam's national Quitline (AAR), in which the Quitline reached out to the patient to engage them in up to 10 sessions of smoking cessation counseling. Retention in Quitline counseling was defined as participating in more than five counseling calls. The study used bivariate and logistic regression analyses to explore the associations between retention and other factors.

### Results

Fifty-one percent of HIV-positive smokers completed more than five counseling sessions. Smokers living with HIV aged 35 or older (OR = 5.53, 95% CI 1.42–21.52), who had a very low/low tobacco dependence level (OR = 2.26, 95% CI 1.14–4.51), had a lower score of

**Funding:** This work was supported by the National Cancer Institute, US National Institutes of Health (grant number R01CA240481). The funders had no role in study design, data collection and analysis, decision to publish, or preparation of the manuscript.

**Competing interests:** The authors have declared that no competing interests exist.

perceived importance of quitting cigarettes (OR = 0.87, 95% CI 0.76–0.99), had a household ban or partial ban on cigarettes smoking (OR = 2.58, 95% CI 1.39–4.80), and had chosen a quit date during the Quitline counseling (OR = 3.0, 95% CI 1.63–5.53) were more likely to retain in the Quitline counseling than those smokers living with HIV whose ages were less than 35, who had a high/very high tobacco dependence level, had a higher score of perception of the importance of quitting cigarettes, did not have a household ban on cigarettes smoking, and did not choose a quit date during counseling.

## Conclusion

There is a high retention rate in Quitline counseling services among PLWHs receiving care at HIV outpatient clinics. Tailoring interventions to the associated factors such as age, tobacco dependence, perceived importance of quitting, household smoking bans, and setting a quit date during counseling may improve engagement and outcomes, aiding in the reduction of smoking prevalence among HIV-positive individuals.

## Introduction

Smoking, a significant global public health issue, has significantly adverse health effects on people living with HIV/AIDS (PLWH) [1]. The prevalence of smoking among HIV-positive populations is substantially higher compared to that of the general population [2–4]. This increases health risks, accelerates the progression of HIV, and complicates HIV management, such as increasing the risks of opportunistic infections, cardiovascular diseases, and cancers among HIV-positive individuals [1, 5, 6]. In Vietnam, a country with a high prevalence of smoking, especially among PLWHs [7], addressing the intersection of smoking and HIV presents unique challenges. Effective smoking cessation interventions are essential for reducing smoking rates and improving the overall health outcomes of PLWHs [8].

Quitline counseling is an effective method for supporting smoking cessation [9]. However, the effectiveness of this intervention relies heavily on the retention of participants throughout the counseling process [10–15]. The rate of smoking abstinence is higher among smokers who receive multiple counseling sessions [10, 16]. Specifically, the quit rate increased from 11% to 14% for proactive Quitline counseling, compared to an increase from 7% to 10% for reactive counseling [16]. Yet retaining people who use tobacco in multiple sessions of counseling remains a challenge due to barriers including lack of trust in the counselors, skepticism about the effectiveness of Quitlines, discomfort with phone counseling, perceived lack of time for counseling, privacy concerns, and issues related to using cell phones [17–19]. People living with HIV may encounter additional obstacles in accessing and maintaining smoking cessation interventions, including stigma, social isolation, economic stressors, HIV-related stress, psychological distress, and competing health priorities [20, 21].

Existing literature indicates several factors that may influence retention in Quitline smoking cessation counseling, including demographic characteristics, smoking history, and psychosocial variables. Factors such as age, gender, socioeconomic status, high levels of nicotine dependence, self-efficacy to quit, and perception of Quitline efficacy have been found to impact retention rates [11, 19, 22–26]. Furthermore, health conditions, comorbidities, mental health status, and social support are associated with retention in Quitline counseling [11, 24,

25, 27]. However, limited research explores retention rates in Quitline counseling and related factors among PLWHs, particularly in low- and middle-income countries like Vietnam.

This study aims to fill this gap by evaluating the factors associated with retention in Quitline counseling for smoking cessation among HIV-positive smokers receiving care at HIV outpatient clinics in Vietnam. Understanding these factors is essential for creating customized interventions that can increase participation and retention in smoking cessation programs, ultimately leading to better health outcomes for HIV-positive individuals. This research will contribute to the global knowledge base on effective smoking cessation strategies within HIV care settings and provide valuable insights for policymakers and healthcare providers in Vietnam and similar contexts.

## Materials and methods

### Data source

The study analyzes data from a randomized controlled trial (RCT, ClinicalTrials.gov ID NCT05162911) that compared the effectiveness of three smoking cessation interventions delivered in 13 HIV OPCs in Ha Noi, Vietnam, from November 2021 to December 2023.

Participants were screened for tobacco use at the time of registration for a routine visit. They were eligible to enroll if they were 18 or older, active patients at the OPCs, current cigarette-only or dual users (waterpipe and cigarettes), had a mobile phone, and lived in Ha Noi. Eligible participants were randomly assigned to one of three smoking cessation interventions: brief counseling + proactive referral to the Vietnam Quitline (Arm 1, Ask, Assist, Refer, AAR), brief counseling + intensive counseling (Arm 2), or brief counseling + intensive counseling + plus NRT (Arm 3).

A total of 221 smokers aged 18 and above living with HIV were enrolled in Arm 1 (AAR) of the RCT. Research indicates that higher connectivity rates are achieved when a proactive approach is used, where the Quitline actively reaches out to patients to engage them in counseling [13]. Therefore, we implemented a proactive referral system in which patients were asked to provide consent for the clinic to share their contact information with the Quitline. Once consent was obtained, the information was transmitted through a secure server to the Quitline, and the Quitline then proactively reached out to the patient to engage them in counseling. The Vietnam Quitline offers up to 10 counseling sessions. The ten calls include five calls within the first month of enrolment and five follow-up calls from months 2–12:the first call right within two days after the patient was referred to the Quitline, the following calls after one week, two weeks, three weeks, four weeks, two months, three months, six months, nine months, and 12 months. On average, the first call lasted 18 minutes, the second to the fifth call lasted 9.3 minutes, and the 6th to the 10th calls lasted 6.8 minutes. For those who did not answer the call, the Quitline would attempt to contact them three times, and after three unsuccessful attempts, they were considered lost to follow-up.

The analysis consisted of baseline survey data from 221 smokers as well as Quitline utilization data. The referral form and Quitline utilization data were used to monitor the reach (% of eligible patients referred and who received at least one call) and fidelity (number and percentage of counseling calls completed, days between the calls, and counseling call time).

Baseline surveys were carried out between December 2022 and June 2023. The survey was conducted in person using a structured questionnaire in Vietnamese language. After eligibility screening for the RCT, eligible participants were asked to participate in a survey before being randomly assigned to the 3 study arms. The survey took approximately 45 minutes, and after its completion, the participant received an incentive of 2.5 USD. All participants were provided written informed consent. This research was approved by the institutional review boards of

the Institute of Social Medical Studies (Decision 08/HDDD-ISMS) and the New York University School of Medicine (ID i19-01783).

## Measures

**Dependent variable.** The median number of completed Quitline counseling sessions was five. We, therefore, defined retention in Quitline counseling as the patient participating in more than five counseling calls from Quitline. The definition of retention in Quitline counseling is in line with previous studies [28].

**Independent variables.** Sociodemographic variables included sex, age, marital status, educational status, household income, employment, and living arrangement.

Health status was measured using a single question with a scale of 1 to 5, where 1 = Poor, 2 = Fair, 3 = Good, 4 = Very Good, and 5 = Excellent [29].

Social support was assessed using the Multidimensional Scale of Perceived Social Support Scale (MPSS) [30], which aggregates three types of social support: significant other, family, and friends. Respondents were asked to rate 12 social support statements on a scale of 1 to 4, where 1 indicated "Strongly disagree" and 4 indicated "Strongly agree." The mean scores for each of the three social support categories were calculated.

Tobacco dependence was assessed using the Fagerstrom Test for Nicotine Dependence, which consists of six items that evaluate the quantity of cigarette consumption, the compulsion to use, and the dependence [31]. The measured levels of tobacco dependence ranged from 'Very low dependence' with a score of 0–2 to 'Very high dependence' with a score of 8–10.

Tobacco use was defined as cigarette-only or dual users who used both cigarettes and water pipes. We also assessed the number of cigarettes smoked per day and the number of times smoked water pipes per day.

Alcohol use was assessed using the Alcohol Use Disorder Identification Test–Consumption (AUDIT–C) [32]. The AUDIT-C scale ranges from 0 to 12. Hazardous drinking was defined with a score of ≥4 for men and ≥3 for women [33].

Drug use was defined as the use of substances for psychotropic rather than medical purposes. We assessed drug use in the past three months. Response options included Opium, Cocaine, Heroin, Amphetamine/Methamphetamine, Marijuana, Ecstasy, MDMA (an abbreviation of 3,4-methylenedioxymethamphetamine), also called "Molly" or "Ecstasy," Ketamine, and others.

Self-efficacy was assessed using the Smoking Abstinence Self-efficacy Questionnaire (SASEQ), which consists of 8 items that evaluated whether the respondent could refrain from smoking in situations on a scale of 1 to 4, where 1 = Not at all sure, 2 = Not very sure, 3 = Fairly sure, and 4 = Absolutely sure. The mean of the sum of scores was calculated [34].

Risk perception was assessed using four questions that asked the respondent about the likelihood of developing illnesses if the respondent continued to smoke on a scale of 1–4, including 1 = Not at all likely, 2 = A little likely, 3 = Very likely, and 4 = Extremely likely. The mean of the sum of scores was calculated [35].

The importance of quitting cigarette smoking was assessed using a single question: "On a scale from 1–10, how important is it to you to quit smoking cigarettes completely?" The scale was 1 to 10, with 1 = not important and 10 = extremely important.

Confidence in quitting cigarettes was assessed using a single question: "On a scale from 1–10, how confident are you that you could quit smoking cigarettes completely or stay quit if you want to?" (1 = not confident at all and 10 = very confident).

Household rules about smoking inside the home were assessed using one question: Is smoking not allowed anywhere, allowed in some places or at times, or allowed everywhere inside the house?

Whether or not patients selected a quit date was documented in the Quitline utilization tracking form. During the first and second counseling sessions, the Quitline counselor asked the smoker to select a quit date on which the smoker would quit cigarettes completely.

## Data analysis

The data were analyzed using Stata (version 14.0). Descriptive statistics were used to summarize the characteristics of PLWHs. Bivariate tests were conducted with a significance level of 0.05. Categorical variables were assessed via chi-square tests, while continuous variables were assessed using t-tests. Multivariable analysis was performed using logistic regression to evaluate the associations between Quitline retention and other patient characteristics. Odds ratios (OR) were reported along with 95% confidence intervals. Independent variables that had a p-value < 0.2 in the bivariate analyses were included in the logistic regression model [36]. P values < 0.05 were considered statistically significant. Multicollinearity was checked using Tolerance and VIF criteria to ensure that the independent variables included in the logistic regression models were not highly correlated with each other.

## Results

### Socio-demographic characteristics of the participants

Most participants were males (95.9%) and over the age of 35 (92.3%). Regarding marital status, 52.9% were married, while 47.1% were single, never married, separated, or divorced. Educational attainment varied, with 43.4% having less than a high school education, 36.2% having completed high school, and 20.4% having vocational training or higher education.

Most participants (90.5%) were employed in salary-paid jobs. In terms of household income, 61.9% had an annual income between VND 100,000,000 and 300,000,000, while 26.2% earned less than VND 100,000,000, and 11.9% earned VND 300,000,000 or more. The majority of patients (70.6%) lived with a spouse/partner or children, 19.9% lived with others, and 9.5% lived alone (Table 2).

### The retention in Quitline smoking cessation counseling and associated factors

Table 1 indicates that a significant majority of participants (87.8%) completed at least one call. More than half (51.1%) completed more than five calls, while a smaller proportion (14.9%) completed all ten calls.

Table 2 shows the results of bivariate analyses that examined the associations between Quitline retention and other factors. Age, number of cigarettes smoked daily, tobacco dependence level, drug use, household smoking bans, and choosing a quit date during counseling were significantly associated with receiving more than five counseling sessions (p<0.05).

**Table 1. Number of Quitline smoking cessation counseling calls received among HIV-positive smokers.**

| Completed Quitline calls (N = 221) | Number of patients | Percentage of completed calls |
|---|---|---|
| no calls | 27 | 12.2 |
| at least 1 call | 194 | 87.8 |
| ≥ 3 calls | 59 | 26.7 |
| ≥ 5 calls | 81 | 36.7 |
| 6 or more calls | 113 | 51.1 |
| 10 calls | 33 | 14.9 |

**Table 2. Factors associated with the number of smoking cessation counseling calls received among HIV-positive smokers.**

| Characteristics (N = 221) | Number of Quitline calls received | | | | | | |
|---|---|---|---|---|---|---|---|
| | Total | | 0–5 calls | | > 5 calls | | p-value |
| | n | %/Mean±SD | n | %/Mean±SD | n | %/Mean±SD | |
| **Gender** | | | | | | | 0.786 |
| Female | 9 | 4.1 | 4 | 44.4 | 5 | 55.6 | |
| Male | 212 | 95.9 | 104 | 49.1 | 108 | 50.9 | |
| **Age** | | | | | | | **0.004** |
| < = 35 | 17 | 7.7 | 14 | 82.4 | 3 | 17.6 | |
| >35 | 204 | 92.3 | 94 | 46.1 | 110 | 53.9 | |
| **Marital status** | | | | | | | 0.163 |
| Married | 117 | 52.9 | 52 | 44.4 | 65 | 55.6 | |
| Single/Never married/Separated/Divorced | 104 | 47.1 | 56 | 53.9 | 48 | 46.1 | |
| **Education** | | | | | | | 0.126 |
| Less than high school | 96 | 43.4 | 49 | 51.0 | 47 | 49.0 | |
| High school | 80 | 36.2 | 43 | 53.8 | 37 | 46.2 | |
| Vocational training/College/University and above | 45 | 20.4 | 16 | 35.6 | 29 | 64.4 | |
| **Occupation** | | | | | | | 0.553 |
| Unemployed/Homemaker | 10 | 4.5 | 6 | 60.0 | 4 | 40.0 | |
| Salary, paid jobs | 200 | 90.5 | 98 | 49.0 | 102 | 51.0 | |
| Other (farmers, Retired/students) | 11 | 5.0 | 4 | 36.4 | 7 | 63.6 | |
| **Household income in the past 12 months** | | | | | | | 0.653 |
| < 100,000,000 | 57 | 26.2 | 25 | 43.9 | 32 | 56.1 | |
| 100,000,000 - < 300,000,000 | 135 | 61.9 | 69 | 51.1 | 66 | 48.9 | |
| > = 300,000,000 | 26 | 11.9 | 13 | 50.0 | 13 | 50.0 | |
| **Living arrangements** | | | | | | | 0.702 |
| Live alone | 21 | 9.5 | 10 | 47.6 | 11 | 52.4 | |
| Live with spouse/partners/children | 156 | 70.6 | 74 | 47.4 | 82 | 52.6 | |
| Live with others | 44 | 19.9 | 24 | 54.6 | 20 | 45.4 | |
| **Type of smoker** | | | | | | | 0.201 |
| Cigarettes only | 111 | 50.2 | 59 | 53.2 | 52 | 46.8 | |
| Dual user | 110 | 49.8 | 49 | 44.5 | 61 | 55.5 | |
| **Number of cigarettes smoked per day** (mean) | 221 | 14.5±8.2 | 108 | 16.1±8.3 | 113 | 13.1±7.7 | **0.006** |
| **Number of times smoking waterpipes per day** (mean) | 110 | 10.5±9.8 | 49 | 9.7±7.5 | 49 | 11.2±11.3 | 0.414 |
| **Tobacco dependence level** | | | | | | | **0.005** |
| High/ very high | 77 | 34.8 | 48 | 62.3 | 29 | 37.7 | |
| Medium | 32 | 14.5 | 17 | 53.1 | 15 | 46.9 | |
| Very low/low | 112 | 50.7 | 43 | 38.4 | 69 | 61.6 | |
| **Drug used** | | | | | | | **0.030** |
| Never | 50 | 22.6 | 21 | 42.0 | 29 | 58.0 | |
| Ever | 131 | 59.3 | 60 | 45.8 | 71 | 54.2 | |
| In the last 3 months | 40 | 18.1 | 27 | 67.5 | 13 | 32.5 | |
| **Hazardous drinking** | | | | | | | 0.099 |
| No | 84 | 38.0 | 47 | 55.9 | 37 | 44.1 | |
| Yes | 137 | 62.0 | 61 | 44.5 | 76 | 55.5 | |
| **Smoking risk perception** (mean) | 221 | 8.2±2.5 | 108 | 8.3±2.5 | 113 | 8.0±2.5 | 0.421 |
| **Self-efficacy** (mean) | 221 | 10.0±5.0 | 108 | 9.4±4.5 | 113 | 10.6±5.4 | 0.072 |
| **Importance of quitting cigarette smoking** (mean) | 221 | 8.4±2.3 | 108 | 8.7±2.2 | 113 | 8.1±2.4 | 0.078 |
| **Confidence in quitting** (mean) | 221 | 6.4±2.6 | 108 | 6.2±2.6 | 113 | 6.7±2.5 | 0.162 |

*(Continued)*

**Table 2.** (Continued)

| Characteristics (N = 221) | Number of Quitline calls received | | | | | | |
|---|---|---|---|---|---|---|---|
| | Total | | 0–5 calls | | > 5 calls | | p-value |
| | n | %/Mean±SD | n | %/Mean±SD | n | %/Mean±SD | |
| **Health status** | | | | | | | 0.145 |
| Good/Very good/Excellent | 76 | 34.4 | 32 | 42.1 | 44 | 57.9 | |
| Fair/Poor | 145 | 65.6 | 76 | 52.4 | 69 | 47.6 | |
| **Social support** | | | | | | | |
| Family support score (mean) | 221 | 3.1±0.5 | 108 | 3.2±0.5 | 113 | 3.1±0.5 | 0.831 |
| Friend support score (mean) | 221 | 2.±0.5 | 108 | 2.±0.5 | 113 | 2.8±0.5 | 0.666 |
| Other support score (mean) | 221 | 3.2±0.5 | 108 | 3.2±0.4 | 113 | 3.1±0.5 | 0.402 |
| **Total social support score** (mean) | 221 | 3.1±0.4 | 108 | 3.1±0.4 | 113 | 3.0±0.4 | 0.537 |
| **Household smoking policy** | | | | | | | **0.001** |
| Smoking is not allowed anywhere/ allowed in some places or at sometimes | 107 | 48.4 | 40 | 37.4 | 67 | 62.6 | |
| Smoking is allowed everywhere inside the home | 114 | 51.6 | 68 | 59.7 | 46 | 40.4 | |
| **Chose a quit date** | | | | | | | **<0.001** |
| No | 97 | 43.9 | 61 | 62.9 | 36 | 37.1 | |
| Yes | 124 | 56.1 | 47 | 37.9 | 77 | 62.1 | |

Note: *Bold values signify significant findings at p<0.05.*

The proportion of receiving more than five smoking cessation counseling calls was higher among HIV-positive smokers who were over 35 years old, who smoked fewer cigarettes daily, those with very low or low tobacco dependence levels, who lived in households with smoking bans, who did not use drugs in the last three months, and had chosen a quit date during counseling compared with those whose age was equal or less than 35, who smoked more cigarettes daily, those with higher tobacco dependence levels, who lived in households without smoking bans, those who had not chosen a quit date during counseling, and who used drugs in the last three months (p<0.05).

Table 3 presents results from logistic regression analyses indicating significant associations between HIV-positive smokers' retention in Quitline counseling (receiving more than five smoking cessation counseling calls) and age, tobacco dependence level, perception of the importance of quitting smoking, household smoking bans, and choosing a quit date during counseling.

The odds of retention in Quitline counseling were significantly higher among HIV-positive smokers who were over the age of 35 (OR = 5.53, 95% CI 1.42–21.52), those with very low or low tobacco dependence levels (OR = 2.26, 95% CI 1.14–4.51), who had a lower score of perceived importance of quitting cigarettes (OR = 0.87, 95% CI 0.76–0.99), those with household smoking bans (OR = 2.58, 95% CI 1.39–4.80), and those who had chosen a quit date during counseling (OR = 3.0, 95% CI 1.63–5.53) compared to those whose age equal or less than 35, those with higher tobacco dependence levels, who had a higher score of perception of the importance of quitting, who did not have household smoking bans, and who did not choose a quit date during counseling.

## Discussion

This is the first study of Quitline retention rates and factors influencing retention in Quitline counseling among PLWHs engaged in care. Overall, the retention rate, defined as receiving more than five completed calls, was over 50%. This retention rate is significantly higher than

**Table 3. Logistic regression of the number of Quitline smoking cessation counseling calls received among HIV-positive smokers.**

| Independent variables (N = 221) | Number of Quitline smoking cessation counseling calls received (0–5 calls vs. > 5 calls) | | |
|---|---|---|---|
| | OR | 95% CI | p-value |
| **Age** | | | |
| < = 35 (*ref.*) | | | |
| >35 | 5.53 | 1.42–21.52 | **0.014** |
| **Tobacco dependence** | | | |
| High/ very high (*ref.*) | | | |
| Medium | 1.43 | 0.55–3.72 | 0.467 |
| Very low/low | 2.26 | 1.14–4.51 | **0.020** |
| **Substance used** | | | |
| Never (*ref.*) | | | |
| Ever | 0.86 | 0.39–1.89 | 0.714 |
| In the last 3 months | 0.40 | 0.14–1.12 | 0.081 |
| **Hazardous drinking** | | | |
| No (*ref.*) | | | |
| Yes | 1.66 | 0.88–3.12 | 0.115 |
| **Self-efficacy score** (Mean/SD) | 1.04 | 0.97–1.10 | 0.269 |
| **Importance of quitting** (Mean/SD) | 0.87 | 0.76–0.99 | **0.046** |
| **Household smoking policy** | | | |
| Smoking is allowed everywhere inside the home (*ref.*) | | | |
| Smoking is not allowed anywhere/ allowed in some places or at sometimes | 2.58 | 1.39–4.80 | **0.003** |
| **Chose a quit date** | | | |
| No (*ref.*) | | | |
| Yes | 3.00 | 1.63–5.53 | **< 0.001** |

Note: *Ref*: *reference group*, OR: *Odds ratio; Bold values signify significant findings at p<0.05. All variables with a P-value <0.2 in the bivariate analyses were included in the logistic regression models.*

previously reported in studies of Quitline retention among a general population of people who use tobacco [10, 28, 37–39]. Quitline retention rates range from 19% to 53%, depending on the study population and definition of retention. For instance, some studies conducted in the US have reported retention rates of 19.0% of smokers completed three or more Quitline counseling sessions [10], 25.6% for participants who received more than five Quitline counseling sessions [28], 30.5%[37] and 38.5% [38] for participants who received five or more Quitline counseling sessions. A study in India found a retention rate of 41.8% of 5.179 participants who received four Quitline counseling calls[39]. A study in Vietnam reported that 64% and 53% of 431 people who smoked and were proactively referred by the health providers at hospitals to the Quitline completed four or more Quitline counseling sessions within the first month from the enrollment, and all eight counseling sessions during 12 months, respectively [15]. Research has found that higher connectivity rates are achieved using the proactive approach evaluated in the current study in which Quitline reached out to patients to engage them in counseling [40]. Studies that have enrolled PLWH have reported low retention rates compared with the current study. For example, 14% of smokers with HIV at OPC clinics in the US received 3–5 Quitline counseling sessions (out of a 5-call counseling program) [41]. The low retention rate of HIV-positive smokers may be attributed to unique challenges that prevent these individuals

from participating and retaining in smoking cessation interventions, such as stigma and psychological distress [20, 21, 42].

The high retention rate in the current study may be related to several characteristics of the intervention and approach to connecting patients to the Quitline. All participants who were identified as current tobacco users were offered brief counseling that was delivered by a trusted health provider in the context of their usual care. Counseling included a discussion about the harms of continuing to smoke, including specific adverse impacts on HIV-related outcomes. Health workers also described the Quitline service and encouraged patients to respond to the calls that they would receive. The finding is in line with previous studies that demonstrate that Ask, Advice, and Connect to Quitline (AAC) has excellent potential to enhance rates of enrollment and retention in Quitline-delivered treatment in primary care settings [10, 40]. The success of this approach emphasizes the importance of customizing cessation intervention programs to serve vulnerable populations better, thereby increasing retention rates and improving overall cessation outcomes.

The study finds several patient characteristics, including older age, lower tobacco dependence, household smoking bans, willingness to set a quit date during initial counseling, and lower perception of the importance of quitting cigarettes, were significantly associated with the higher rates of retention of engaging in more than five sessions of Quitline counseling. The strong link between older age and higher retention rates may be attributed to the greater awareness and concern for health often seen in older adults. Older individuals are more conscious of the health consequences of smoking, such as heart disease, respiratory issues, and cancer, as well as the benefits of quitting [43]. This heightened awareness motivates them to adhere to smoking cessation interventions and increases the likelihood of successfully quitting. This finding is supported by other studies indicating that older smokers are more likely to participate in and benefit from smoking cessation programs [11, 22, 23].

Higher retention rates were also associated with lower levels of tobacco dependence. People with lower nicotine dependence experience fewer withdrawal symptoms, which makes quitting smoking less challenging and more manageable. Lower dependence reduces the physical and psychological barriers to quitting, and this may result in greater willingness to continue in treatment. This finding is in lines with existing literature, which suggest that the severity of nicotine dependence can affect both the motivation to quit and the ability to maintain abstinence [23, 44–47].

The presence of household smoking bans was significantly associated with retention rates among HIV-positive smokers, indicating the critical role of the social environment in smoking cessation efforts. Supportive home environments where smoking is restricted can create a conducive atmosphere for smokers attempting to quit, reducing their exposure to triggers and encouraging a smoke-free lifestyle. This finding aligns with studies showing that household smoking bans can significantly enhance the effectiveness of smoking cessation interventions by providing a supportive environment for individuals attempting to quit smoking [48]. Given these findings, it is recommended that smoking cessation programs actively promote the implementation of household smoking bans as part of their intervention strategies. By doing so, programs can leverage the positive impact of a supportive home environment to improve retention rates and overall success in quitting smoking.

Setting a quit date was another critical factor associated with better retention. This proactive step likely reflects a higher level of commitment and readiness to quit smoking, which translates to greater engagement with counseling services. Setting a quit date provides a clear, actionable target for smokers, helping to transform abstract intentions into concrete plans. Moreover, the process of setting a quit date can serve as a psychological commitment, reinforcing the smoker's intention to quit. Studies examining Quitline callers and smokers receiving

smoking cessation counseling found that high-quality quit-date goal setting was significantly associated with higher odds of making a quit attempt and maintaining abstinence [49, 50]. Individuals who establish a quit date within the first two weeks of starting a cessation program show significantly higher success rates [49–51]. It is recommended that Quitlines and other smoking cessation programs emphasize the importance of setting a quit date during initial counseling sessions.

A lower perception of the importance of quitting was associated with higher retention in Quitline counseling. This finding contradicts findings from previous studies that a higher perception of the importance of quitting is associated with higher retention in smoking cessation interventions. This counterintuitive finding suggests that smokers who do not initially recognize the importance of quitting may still engage with counseling, possibly due to external motivations such as social support, family pressure, or healthcare provider recommendations. Research indicates that even if initial motivation is low, the structured support and continuous engagement provided by smoking cessation programs can foster a gradual shift in smokers' attitudes toward quitting [48]. Moreover, external motivations such as social support from family and friends or encouragement from healthcare providers are important in keeping individuals engaged in cessation efforts [19, 52].

The study provides valuable insights into smoking cessation programs and the factors influencing retention in Quitline counseling. However, several limitations should be considered when interpreting the results. The participants were drawn from a sample of PLWHs receiving treatment at HIV OPCs, which may not fully represent the broader population of PLWHs in Vietnam. It is important to note, however, that the majority of PLWH in Vietnam receive antiretroviral therapy (ART) at OPCs. The study sample mostly consisted of men (95.9%) aged 35 and above (92.3%), which limited the generalizability of the findings to gender and age. The high proportion of men is due to the study's focus on recruiting HIV-positive patients who smoke. Since the rate of smoking among men was high (45.3%) and among women in Vietnam was very low (1.1%) [53], very few eligible female patients were available for recruitment at the OPCs. The majority of participants were over 35 years old, possibly due to the recent decrease in new HIV infections among young people and the declining smoking rates among younger individuals in Vietnam [53]. As a result, fewer young HIV-positive patients who smoked were recruited for the study at the OPCs. There may have been significant variation in how brief counseling and the referral process were implemented across the OPCs enrolled in the study. However, protocols and training were standardized, and healthcare workers received a one-page coaching guide to support brief counseling and Quitline referral conversations [54]. The study also relied on self-reported data to explore health behaviors, which may be subject to recall bias, reporting inaccuracies, and social desirability. This means that participants might have underreported their smoking habits and behaviors, impacting the reliability of the findings.

## Conclusion

This study identifies several key factors related to retention in Quitline counseling among people living with HIV (PLWH) who use tobacco in Vietnam, including age, tobacco dependence, household smoking bans, perceived importance of quitting, and setting a quit date during counseling. These findings can help in developing tailored smoking cessation strategies for this population. Similar to previous research, a proactive approach resulted in high retention rates among PLWHs receiving care in HIV outpatient clinics. Integrating tobacco cessation counseling within HIV care, along with proactive linkage of patients to national Quitlines in low- and middle-income countries (LMICs), is a promising strategy for improving access to treatment.

Future research should explore additional psychosocial and structural barriers to retention in Quitline counseling, particularly in low- and middle-income countries, to further refine and enhance smoking cessation interventions within HIV care settings aiming to improve the health of PLWHs.

## Supporting information

**S1 Dataset.**
(DTA)

## Author Contributions

**Conceptualization:** Nam Truong Nguyen, Donna Shelley.

**Data curation:** Trang Nguyen, Charles M. Cleland, Yen Pham, Nga Truong, Reet Kapur, Phuong Thu Phan.

**Formal analysis:** Nam Truong Nguyen, Trang Nguyen, Giap Van Vu, Charles M. Cleland, Yen Pham, Nga Truong, Reet Kapur, Gloria Guevara Alvarez, Phuong Thu Phan, Mari Armstrong-Hough, Donna Shelley.

**Funding acquisition:** Nam Truong Nguyen, Donna Shelley.

**Investigation:** Nam Truong Nguyen, Trang Nguyen, Giap Van Vu, Charles M. Cleland, Yen Pham, Nga Truong, Reet Kapur, Gloria Guevara Alvarez, Phuong Thu Phan, Mari Armstrong-Hough, Donna Shelley.

**Methodology:** Nam Truong Nguyen, Trang Nguyen, Donna Shelley.

**Resources:** Trang Nguyen, Giap Van Vu, Charles M. Cleland, Yen Pham, Nga Truong, Reet Kapur, Gloria Guevara Alvarez, Phuong Thu Phan.

**Validation:** Nam Truong Nguyen, Trang Nguyen, Giap Van Vu, Charles M. Cleland, Yen Pham, Nga Truong, Reet Kapur, Gloria Guevara Alvarez, Phuong Thu Phan, Mari Armstrong-Hough, Donna Shelley.

**Visualization:** Nga Truong.

**Writing – original draft:** Nam Truong Nguyen.

**Writing – review & editing:** Nam Truong Nguyen, Trang Nguyen, Giap Van Vu, Charles M. Cleland, Yen Pham, Nga Truong, Reet Kapur, Gloria Guevara Alvarez, Phuong Thu Phan, Mari Armstrong-Hough, Donna Shelley.

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
