## [Decision Letter · Decision Letter 0]

1 Oct 2024

PONE-D-24-35771Factors Associated with Retention in Quitline Counseling for Smoking Cessation among HIV-Positive Smokers Receiving Care at HIV Outpatient Clinics in VietnamPLOS ONE

Dear Dr. Nguyen,

Thank you for submitting your manuscript to PLOS ONE. After careful consideration, we feel that it has merit but does not fully meet PLOS ONE’s publication criteria as it currently stands. Therefore, we invite you to submit a revised version of the manuscript that addresses the points raised during the review process.

We look forward to receiving your revised manuscript.

Kind regards,

Ching Sin Siau

Academic Editor

PLOS ONE

Journal Requirements: When submitting your revision, we need you to address these additional requirements. 1. Please ensure that your manuscript meets PLOS ONE's style requirements, including those for file naming. The PLOS ONE style templates can be found at https://journals.plos.org/plosone/s/file?id=wjVg/PLOSOne_formatting_sample_main_body.pdf and https://journals.plos.org/plosone/s/file?id=ba62/PLOSOne_formatting_sample_title_authors_affiliations.pdf 2. Please ensure you have included the registration number for the clinical trial referenced in the manuscript. 3. Thank you for stating the following financial disclosure: "This work was supported by the National Cancer Institute, US National Institutes of Health (grant number R01CA240481)." Please state what role the funders took in the study.  If the funders had no role, please state: ""The funders had no role in study design, data collection and analysis, decision to publish, or preparation of the manuscript."" If this statement is not correct you must amend it as needed. Please include this amended Role of Funder statement in your cover letter; we will change the online submission form on your behalf. 4. We note that your Data Availability Statement is currently as follows: All relevant data are within the manuscript and its Supporting Information files. Please confirm at this time whether or not your submission contains all raw data required to replicate the results of your study. Authors must share the “minimal data set” for their submission. PLOS defines the minimal data set to consist of the data required to replicate all study findings reported in the article, as well as related metadata and methods (https://journals.plos.org/plosone/s/data-availability#loc-minimal-data-set-definition). For example, authors should submit the following data: - The values behind the means, standard deviations and other measures reported;- The values used to build graphs;- The points extracted from images for analysis. Authors do not need to submit their entire data set if only a portion of the data was used in the reported study. If your submission does not contain these data, please either upload them as Supporting Information files or deposit them to a stable, public repository and provide us with the relevant URLs, DOIs, or accession numbers. For a list of recommended repositories, please see https://journals.plos.org/plosone/s/recommended-repositories. If there are ethical or legal restrictions on sharing a de-identified data set, please explain them in detail (e.g., data contain potentially sensitive information, data are owned by a third-party organization, etc.) and who has imposed them (e.g., an ethics committee). Please also provide contact information for a data access committee, ethics committee, or other institutional body to which data requests may be sent. If data are owned by a third party, please indicate how others may request data access. 5. Please review your reference list to ensure that it is complete and correct. If you have cited papers that have been retracted, please include the rationale for doing so in the manuscript text, or remove these references and replace them with relevant current references. Any changes to the reference list should be mentioned in the rebuttal letter that accompanies your revised manuscript. If you need to cite a retracted article, indicate the article’s retracted status in the References list and also include a citation and full reference for the retraction notice.

Reviewers' comments:

Reviewer's Responses to Questions

**Comments to the Author**

1. Is the manuscript technically sound, and do the data support the conclusions?

Reviewer #1: Yes

Reviewer #2: Yes

2. Has the statistical analysis been performed appropriately and rigorously? 

Reviewer #1: Yes

Reviewer #2: Yes

3. Have the authors made all data underlying the findings in their manuscript fully available?

Reviewer #1: Yes

Reviewer #2: Yes

4. Is the manuscript presented in an intelligible fashion and written in standard English?

Reviewer #1: Yes

Reviewer #2: Yes

5. Review Comments to the Author

Reviewer #1: Well written article with good command of the English language. Statistical analysis were also very clear and done well. Only a few grammatical error that needs edition. Otherwise, it is a good study to be published.

Reviewer #2: This study addresses a critical gap in smoking cessation efforts among HIV-positive populations, particularly in low- and middle-income countries like Vietnam. The focus on retention in Quitline counseling is especially important, as sustained engagement in smoking cessation programs is key to their success. The study’s findings provide valuable insights into the demographic factors associated with retention, potentially informing the design of tailored interventions for HIV-positive smokers. Additionally, it contributes to the growing body of knowledge on improving the effectiveness of Quitline services in a vulnerable population that often faces complex health challenges.

Comments to Author:

Abstract: “Proactive referral to Quitline resulted in high rates of retention in Quitline counseling services among PLWHs receiving care at HIV outpatient clinics.” – The study does not compare the retention rates of participants who received proactive referrals with those who did not. The authors are encouraged to revise this statement.

Methodology: To better understand the implementation of the Quitline counseling intervention, it would be helpful if the authors provided more detailed information about the specific protocol used, such as the frequency and duration of calls, the number of follow-up attempts made, and how "lost to follow-up" was determined.

Results:

(a) There is concern about potential limitations related to participant demographics. With ~96% male and ~92% of participants over the age of 35, the sample lacks diversity in terms of gender and age. This could lead to biased results and limit the generalizability of the findings to younger smokers or females living with HIV, who may have different behaviors and challenges when engaging with smoking cessation services. The authors should discuss this limitation in the article. What is the prevalence of smoking among males and individuals aged 35 and above in Vietnam?

(b) While the study reports that age is associated with retention in Quitline counseling, with ~92% of participants over the age of 35, the authors reported through bivariate and multivariate analysis that participants over 35 have higher retention rates. It would be helpful to also report the effect size to demonstrate the relevance of these findings.

6. PLOS authors have the option to publish the peer review history of their article (what does this mean?). If published, this will include your full peer review and any attached files.

Reviewer #1: No

Reviewer #2: No

---

## [Author Response · Author response to Decision Letter 0]

7 Oct 2024

Response to Reviewers and Editor for the manuscript "Factors Associated with Retention in Quitline Counseling for Smoking Cessation among HIV-Positive Smokers Receiving Care at HIV Outpatient Clinics in Vietnam"

Dear PLOS ONE Editors and Reviewers,

We would like to extend our sincere thanks to the editor and reviewers for their valuable feedback and insightful comments. These suggestions have been instrumental in improving the quality of our manuscript.

We have carefully addressed all the comments and made corresponding revisions to the manuscript. We believe that these changes have significantly strengthened the manuscript. The revised manuscript is submitted alongside this response letter.

Below, we have provided a detailed explanation of how we have addressed each of the editor’s and reviewers' comments. We have highlighted our responses to each requirement, starting with the word “Response”.

Thank you once again for your time and consideration.

A. Journal Requirements

Response:

I have revised and reformatted the headings in both the abstract and the manuscript (levels 1, 2, and 3) to ensure they meet PLOS ONE's style requirements. Additionally, I have reviewed and revised the format of the tables to confirm they are consistent with PLOS ONE's guidelines. File naming conventions have also been checked and updated where necessary to comply with the journal's submission standards.

2. Please ensure you have included the registration number for the clinical trial referenced in the manuscript.

Response:

Thank you for your comment regarding the inclusion of the clinical trial registration number. I have ensured that the registration number is included in the manuscript. It has been added to the data source section within the methods section, which now reads:

"Data source 

The study analyzes data from a randomized controlled trial (RCT, ClinicalTrials.gov ID NCT05162911) that compared the effectiveness of three smoking cessation interventions delivered in 13 HIV OPCs in Ha Noi, Vietnam, from November 2021 to December 2023."

3. Thank you for stating the following financial disclosure: "This work was supported by the National Cancer Institute, US National Institutes of Health (grant number R01CA240481)."

Response:

We have received funding from the National Cancer Institute, US National Institutes of Health (grant number R01CA240481). The funder had no role in the study as stated “The funders had no role in the study design, data collection and analysis, decision to publish, or preparation of the manuscript”. I have included this amended Role of Funder statement in our cover letter, as requested.

Please confirm at this time whether or not your submission contains all raw data required to replicate the results of your study. Authors must share the “minimal data set” for their submission. PLOS defines the minimal data set to consist of the data required to replicate all study findings reported in the article, as well as related metadata and methods.

Response:

Following the journal's requirements, we have made the data fully available. We have uploaded a minimal anonymized data set as a Supporting Information file. This data set includes all the necessary data to replicate the study findings reported in the article, as well as the related methods.

Response:

I have carefully reviewed and revised the reference list to ensure it is complete and accurate. Several citations in the main text were updated or corrected to align with the references. The specific revisions are as follows:

• Citation 16: Revised cited texts in lines 4-5 of the 2nd paragraph in the Introduction section.

• Citation 39: Corrected the cited percentage in line 10 of the 1st paragraph in the Discussion section.

Additionally, I have replaced Reference #29 with a more appropriate source. Reference #29, which was cited in the 2nd paragraph of the "Independent Variables, Measures" section in the Methods, has been updated as follows:

• Removed reference:

Herdman M, Gudex C, Lloyd A, Janssen M, Kind P, Parkin D, et al. Development and preliminary testing of the new five-level version of EQ-5D (EQ-5D-5L). Quality of life research. 2011;20(10):1727-36.

• Replaced with:

Ware JE, Jr., Sherbourne CD. The MOS 36-item short-form health survey (SF-36). I. Conceptual framework and item selection. Med Care. 1992;30(6):473-83.

These changes have been reported in the rebuttal letter accompanying our revised manuscript.

B. Responses to reviewer’s comments

Reviewer #1: Well written article with good command of the English language. Statistical analysis were also very clear and done well. Only a few grammatical error that needs edition. Otherwise, it is a good study to be published.

Response:

Thank you for your encouraging words about our manuscript and valuable comments regarding the corrections to the English grammar. I have carefully reviewed and made the necessary corrections in the Material and Methods, Results, and Discussion sections based on your suggestions.

1. Material and Methods:

o Data Source: In the 5th paragraph, I have added the missing word "language" in "Vietnamese language" (2nd line).

o I have also added the word "were" in "All participants were provided..." (5th line).

2. Results:

o Socio-demographic characteristics: In the 1st paragraph, I have corrected "Most participants were male" to "Most participants were males" to reflect the plural form.

3. Discussion:

o In the 4th paragraph, I have changed "rate" to "rates" (4th line) to reflect the plural form.

Reviewer #2: This study addresses a critical gap in smoking cessation efforts among HIV-positive populations, particularly in low- and middle-income countries like Vietnam. The focus on retention in Quitline counseling is especially important, as sustained engagement in smoking cessation programs is key to their success. The study’s findings provide valuable insights into the demographic factors associated with retention, potentially informing the design of tailored interventions for HIV-positive smokers. Additionally, it contributes to the growing body of knowledge on improving the effectiveness of Quitline services in a vulnerable population that often faces complex health challenges.

Comments to Author:

Reviewer #2 Abstract: “Proactive referral to Quitline resulted in high rates of retention in Quitline counseling services among PLWHs receiving care at HIV outpatient clinics.” – The study does not compare the retention rates of participants who received proactive referrals with those who did not. The authors are encouraged to revise this statement.

Response:

Thank you for your comment regarding the statement in the abstract conclusion. We have revised the statement to ensure it does not imply a comparison between participants who received proactive referrals and those who did not. The updated statement now reads: "Proactive referral to Quitline was associated with high retention rates in Quitline counseling services among PLWHs receiving care at HIV outpatient clinics." This revised wording more accurately reflects the study's findings without suggesting a direct comparison.

Reviewer #2. Methodology: To better understand the implementation of the Quitline counseling intervention, it would be helpful if the authors provided more detailed information about the specific protocol used, such as the frequency and duration of calls, the number of follow-up attempts made, and how "lost to follow-up" was determined.

Response:

Thank you for your comment regarding the need for more detailed information on the Quitline counseling intervention. In response, we have added specific details about the Quitline protocol to the method section. The revised text now includes the following information: "The ten calls include five calls within the first month of enrollment and five follow-up calls from months 2-12: the first call right within two days after the patient was referred to the Quitline, the following calls after one week, two weeks, three weeks, four weeks, two months, three months, six months, nine months, and 12 months. On average, the first call lasted 18 minutes, the second to the fifth call lasted 9.3 minutes, and the 6th to the 10th calls lasted 6.8 minutes. For those who did not answer the call, the Quitline would attempt to contact them three times, and after three unsuccessful attempts, they were considered lost to follow-up."

Reviewer #2. Results:

(a) There is concern about potential limitations related to participant demographics. With ~96% male and ~92% of participants over the age of 35, the sample lacks diversity in terms of gender and age. This could lead to biased results and limit the generalizability of the findings to younger smokers or females living with HIV, who may have different behaviors and challenges when engaging with smoking cessation services. The authors should discuss this limitation in the article. What is the prevalence of smoking among males and individuals aged 35 and above in Vietnam?

Response:

Thank you for your comment. I have added a discussion of this limitation in the discussion section of the manuscript:

“The study sample mostly consisted of men (95.9%) aged 35 and above (92.3%), which limited the generalizability of the findings to gender and age. The high proportion of men is due to the study's focus on recruiting HIV-positive patients who smoke. Since the rate of smoking among men was high (45.3%) and among women in Vietnam was very low (1.1%), very few eligible female patients were available for recruitment at the OPCs. The majority of participants were over 35 years old, possibly due to the recent decrease in new HIV infections among young people and the declining smoking rates among younger individuals in Vietnam. As a result, fewer young HIV-positive patients who smoked were recruited for the study at the OPCs.”

Regarding smoking prevalence, the smoking rate among males in Vietnam is 45.3% (MOH, WHO. Global Tobacco Survey Vietnam 2015 - GATS 2015). While specific data for individuals aged 35 and above were not available, the smoking rates among people aged 25-44, 45-64, and 65+ are reported as 27%, 26.9%, and 14.9%, respectively (GATS 2015).

Reviewer #2. (b) While the study reports that age is associated with retention in Quitline counseling, with ~92% of participants over the age of 35, the authors reported through bivariate and multivariate analysis that participants over 35 have higher retention rates. It would be helpful to also report the effect size to demonstrate the relevance of these findings.

Response:

Thank you for your insightful comment regarding the need for reporting an effect size to demonstrate the relevance of the association between age and retention in Quitline counseling. In our bivariate analyses, we examined the association between age and retention (measured by the number of Quitline calls received). A chi-square test was conducted to assess this relationship, and the results indicated a statistically significant association between age and retention (p < 0.05, χ² = 8.26). Specifically, the p-value of 0.004, as shown in Table 2, confirms the significance of the association.

In our multiple logistic regression analysis, we have already reported the odds ratio (OR) and its 95% confidence interval (CI) for the age variable, which serves as the appropriate effect size in the context of logistic regression. The odds ratio (OR) of 5.53 (95% CI: 1.42–21.52) indicates that participants over the age of 35 were over five times more likely to be retained in Quitline counseling compared to younger participants, demonstrating the magnitude and practical significance of this association. The odds ratio and confidence interval are presented in Table 3 in the manuscript, highlighting the statistically significant association between age and retention.

Additionally, to further illustrate the importance of age in predicting retention, we conducted a likelihood ratio test comparing two models: one with age as an independent variable and one without. The test showed a significant improvement in the model's fit with the inclusion of the age variable (Log-likelihood ratio chi-squared = 7.5, P = .023). This result further underscores that age is a meaningful predictor of retention in the Quitline counseling program.

---

## [Decision Letter · Decision Letter 1]

9 Dec 2024

Factors Associated with Retention in Quitline Counseling for Smoking Cessation among HIV-Positive Smokers Receiving Care at HIV Outpatient Clinics in Vietnam

PONE-D-24-35771R1

Dear Dr. Nguyen,

We’re pleased to inform you that your manuscript has been judged scientifically suitable for publication and will be formally accepted for publication once it meets all outstanding technical requirements.

Kind regards,

Ching Sin Siau

Academic Editor

PLOS ONE

Additional Editor Comments (optional):

In the abstract, you have written "Proactive referral to Quitline was associated with high retention rates...". Please consider changing this sentence to "There is a high retention rate in Quitline counseling services among PLWHs receiving care at HIV outpatient clinics." This is to more accurately reflect that this is a descriptive observation, rather than an observation derived from inferential statistical analysis.

Reviewers' comments:

Reviewer's Responses to Questions

**Comments to the Author**

1. If the authors have adequately addressed your comments raised in a previous round of review and you feel that this manuscript is now acceptable for publication, you may indicate that here to bypass the “Comments to the Author” section, enter your conflict of interest statement in the “Confidential to Editor” section, and submit your "Accept" recommendation.

Reviewer #1: All comments have been addressed

Reviewer #2: All comments have been addressed

2. Is the manuscript technically sound, and do the data support the conclusions?

Reviewer #1: Yes

Reviewer #2: Yes

3. Has the statistical analysis been performed appropriately and rigorously? 

Reviewer #1: Yes

Reviewer #2: Yes

4. Have the authors made all data underlying the findings in their manuscript fully available?

Reviewer #1: Yes

Reviewer #2: Yes

5. Is the manuscript presented in an intelligible fashion and written in standard English?

Reviewer #1: Yes

Reviewer #2: Yes

6. Review Comments to the Author

Reviewer #1: (No Response)

Reviewer #2: (No Response)

7. PLOS authors have the option to publish the peer review history of their article (what does this mean?). If published, this will include your full peer review and any attached files.

Reviewer #1: **Yes: **Dr Subashini A/P Ambigapathy

Reviewer #2: No

---

## [Editor Report · Acceptance letter]

16 Dec 2024

PONE-D-24-35771R1 

PLOS ONE

Dear Dr. Nguyen, 

I'm pleased to inform you that your manuscript has been deemed suitable for publication in PLOS ONE. Congratulations! Your manuscript is now being handed over to our production team.

Kind regards, 

on behalf of

Dr. Ching Sin Siau 

Academic Editor

PLOS ONE